# Exploring Factors Linked to the Mathematics Achievement of Ethnic Minority Students in China for Sustainable Development: A Multilevel Modeling Analysis

**Aoxue Su \* and Wei He**

College of Science, Minzu University of China, Beijing 100081, China; 9900564@muc.edu.cn
\* Correspondence: suaoxue@muc.edu.cn; Tel.: +86-10-68933910

**Abstract:** To improve the sustainable development of minority education and ensure equitable quality education, this study explored student- and school-related factors linked to the mathematics achievement of minority senior high school students in China. Based on the data obtained from 932 teachers and 1873 students, within 31 interior ethnic boarding schools in 14 provinces of China, multilevel analysis showed that gender, class organization, learning strategies, and learning self-efficacy were significant student-level predictors of mathematics achievement. Students were more likely to score highly if they were boys, were in mixed classes, had more self-efficacy in learning mathematics, and used more effective mathematics learning strategies. At the school level, teachers' job satisfaction positively predicted students' mathematics achievement. Additionally, there was a significant interaction between school location and expected class organization in relationship to students' mathematics achievement. For schools located in the urban center, the effect of class organization on students' mathematics achievement was greater than schools located in the suburbs. For the sustainable development of minority education, it is necessary to further promote mixed-class teaching, set such schools in the suburbs, and improve teachers' job satisfaction through multiple measures.

**Keywords:** interior ethnic minority boarding schools; sustainable development; equitable quality education; mathematics achievement; multilevel modeling

## 1. Introduction

Most countries across the world are multiethnic states, and most Indigenous people and minorities are in weak positions both politically and economically. Education plays an important role in promoting the integration of ethnic minorities into mainstream society. However, a large amount of empirical evidence has highlighted that ethnic minority students often face negative outcomes associated with schooling in comparison to their mainstream counterparts [1], including poor academic achievement, lower enrollment and graduation rates, and higher school dropout rates. Various education policies have been formulated to promote education equity and guarantee the right to education for ethnic minorities. For example, the landmark No Child Left Behind Act and Every Student Succeeds Act in the United States; Indigenous Education (Target Assistant) Act 2000 in Australia; and Ka Hikitia—Managing for Success in New Zealand. On the basis of such policies, a number of more specific measures have been implemented. Academically rich boarding schools provide access to better education to minority students. For example, public boarding schools like the Schools for Educational Evolution and Development (SEED) have been established in the United States for disadvantaged students [2]. In Australia, boarding school models were recommended by the 2014 Wilson review of

Indigenous Education in the Northern Territory as the preferred secondary education option for very remote Aboriginal students [3].

China is a multinational state with 56 minority groups and more than 100 million minority populations. Education, especially ethnic minority education, has taken a great leap forward since China's reform and opening up (1978 to present), and the education levels of the minority population have increased dramatically [4]. Take Uighurs (a Muslim ethnic minority in northwestern China) as an example. According to the fifth and sixth national census, from 2000 to 2010, the percentages of ethnic Uighurs who received a junior high school education and senior high school education increased from 24.6% to 42.0% and from 4.3% to 6.6%, respectively. Despite the remarkable gains, ethnic minority education still faces some challenges. For example, statistics compiled in 2010 indicate that 6.3% of the Uyghur sample population has a college education level, compared to 9.7% in the Han (dominant nationality of China) sample populations [5].

A vast majority of China's minority populations live in the western and frontier areas of China. Due to historical, natural, and geographical conditions, as well as other factors, the educational foundations of ethnic areas are still very weak, and there is still a considerable gap in educational resources and professional teachers compared with the developed inland areas. As one of the sustainable development goals set by the United Nations, access to "inclusive and equitable quality education" is vital for improving people's lives and sustainable development [6]. Therefore, to ensure equitable quality education, the Chinese government decided to further adopt preferential policies to increase intellectual aid given to ethnic minorities, including the interior ethnic boarding schooling policy [7,8]. Since 1984, the Chinese government has founded interior ethnic boarding schools for students from Tibet. Through this project, Tibetan primary school graduates are recruited each year to complete secondary school and university studies in developed eastern cities of China. Drawing on these experiences, interior ethnic boarding schools have been founded in developed inland cities for students from Xinjiang since 2000.

This special schooling policy aims to provide ethnic minority students with a more complete education, with better educational resources and more professional teachers than those that may be available at schools in remote ethnic areas. This special schooling policy has been implemented for 35 years, and a cumulative total of more than a quarter of a million ethnic minority students have been enrolled. Has the goal of improving the quality of minority education been achieved? Does this schooling policy need to be adjusted? The answers to these questions should be given on the basis of empirical research.

Students' academic achievement is an important measure of the quality of education. To improve students' academic achievement, it is necessary to identify factors linked to students' academic achievement. The most influential large-scale educational assessments in the world, such as the Programme for International Student Assessment (PISA), Trends in International Mathematics and Science Study (TIMSS), and Progress in International Reading Literacy Study (PIRIS), also provide detailed analyses of factors that influence students' academic achievement [9,10]. School is universally the main institute for teaching and learning, so school-related factors (e.g., school characteristics and school climate) are possible predictors of students' academic achievement. Additionally, students bring their own traits (e.g., motivation) and background characteristics (e.g., socioeconomic status) when they attend school. Therefore, student- and school-related factors are the main predictors of students' academic achievement.

However, factors that affect academic achievement vary according to cultural context. Therefore, based on the practical experience of large-scale educational assessments and the characteristics of these ethnic minority students, multilevel modeling analysis was performed to explore student- and school-related factors linked to students' academic achievement. The findings could be of great significance to the improvement of educational decision-making and school management, and the ultimate realization of quality-balanced development of education in China.

## 2. Literature Review

### 2.1. Research on the Academic Status of Ethnic Minority Students in China

Studies have highlighted that "science and engineering problems" commonly exist in minority education in China; that is, ethnic minority students in basic education have prominent learning difficulties in mathematics and other science subjects. This, as well as a poor quality of science teaching, leads to students generally abandoning science and choosing liberal arts after they have gone to senior high schools and universities. The "science and engineering problem" leads to the unreasonable professional structure of minority talents training, which makes minority college graduates face more severe employment difficulties [11]. The quality of science education is related to the cognitive development and employment prospects of minority students, and social harmony and stability [12]. Moreover, previous studies have found that senior high school students' scores in physics, chemistry, and other science subjects are significantly correlated with their mathematics scores. The correlation between mathematics scores and the total score is also the highest; that is, mathematics occupies a dominant position [13–15]. Therefore, the quality of mathematics education is of great significance to the future career development of minority students and the sustainable development of minority areas in China.

Compared with mainstream students, ethnic minority students in interior ethnic minority boarding schools often exhibit lower academic achievement, especially in mathematics [16]. Reasons for the lower mathematics achievement of these ethnic minority students are mainly related to knowledge foundation, learning methods, and learning motivation. Due to the shortage of qualified teachers and teaching resources in ethnic areas, it is a common phenomenon that ethnic minority students have a weak learning foundation. In addition, the teaching method of teachers in ethnic areas is mainly based on the traditional lecture-style teaching method, which relies on language output and memory learning. As a result, ethnic minority students generally adopt learning methods of passive listening and mechanical memorization, with a lack of interactive communication, independent thinking, and exploration [17]. Furthermore, these ethnic minority students are excellent junior high school graduates selected in ethnic areas through a unified admission examination, and therefore usually have high expectations of learning. However, there is still a big academic gap between them and local students. The fierce competition in learning and the psychological gap affect the learning confidence and motivation of these minority students [18].

### 2.2. Research on Student-Level Factors Associated with Students' Academic Achievement

Empirical evidence has shown that students' background characteristics (e.g., gender and socioeconomic status) are significantly related to their academic achievement. Gender differences in the mathematic achievement of high school students have not reached a consistent conclusion so far [19]. Most studies have concluded that boys generally perform better on mathematics tests than girls, but some also shown that gender differences in mathematics achievement either do not exist or favor female students [20]. Therefore, gender differences in mathematics achievement are not universal, and may be linked to one's cultural background.

Socioeconomic status (SES) has long been used to explain differences in students' academic achievement. From the early influential American Coleman report and the UK Plowden report to the international large-scale academic achievement tests such as PISA and TIMSS, the fact that SES significantly affects students' academic achievement has become a consensus [21,22]. However, some researchers have presented different conclusions. For example, by comparing research from different countries, Heyneman and Loxley found that the academic achievement of students in underdeveloped countries had no significant relationship with students' family background, but rather is related to the less obvious social stratification in these countries [23].

As well as students' background characteristics, non-cognitive factors, such as learning strategies and learning self-efficacy, are also reported to relate to students' mathematics achievement. "Learn

to learn" has been regarded as the core of education in the 21st century by UNESCO. The application of learning strategies, as the concrete embodiment of "learn to how to learn", is the basis of one's self-learning ability and provides the possibility for lifelong learning. Numerous empirical studies have highlighted that learning strategies have a significant correlation with students' academic achievements [24,25]. Moreover, findings have also shown that learning self-efficacy and academic achievement are closely interrelated [26,27]. Specifically, it is worth noting that there is a big-fish–little-pond effect in learning self-efficacy. That is, for equally able students, students in high-ability schools or classes have a lower level of learning self-efficacy than those in lower-ability schools or classes, so are likely to experience a decline in academic achievement [28,29].

## 2.3. Research on School-Level Factors Associated with Students' Academic Achievement

Since Coleman et al. began to explore school effects on students' academic achievement [21], research on the effects of school-related factors (school characteristics and school spirit) on students' academic achievement formed a large part of educational research in the twentieth century. Empirical evidence has shown that school characteristics are significantly related to their students' academic achievement. For example, findings have consistently shown that school location is significantly related to students' academic achievement. Students from rural schools often have a lower level of family SES and place less value on academics, and rural schools often have a shortage of school resources. As a result, students from rural schools often lag behind their urban counterparts in terms of academic achievement [30].

At the same time, studies have shown that good discipline and a strong learning atmosphere will help schools carry out teaching activities. The school spirit not only includes the overall code of conduct and values of the school, but also the relationship between teachers and students, peer relations, and the learning atmosphere. According to the OECD's review, among the effective characteristics of a school, the most important index is school spirit [31].

Teachers' job satisfaction has been recognized as important evidence for measuring the effectiveness of school management. Teachers' job satisfaction is a subjective value judgment by teachers, which includes not only teachers' internal satisfaction with self-expectation and self-realization, but also teachers' external satisfaction with working conditions, the working environment, the salary, and other aspects [32]. Findings have consistently shown that teachers' job satisfaction is significantly correlated with students' academic achievement; that is, teachers with higher satisfaction can bring better changes to students and schools, such as improving teaching quality and students' academic achievement [33].

## 2.4. Research on Cross-level Interaction between Student- and School-level Factors

When exploring factors linked to students' academic achievement using multilevel modeling analysis, cross-level interaction between student- and school-level factors could be also analyzed, to achieve a comprehensive understanding of the factors linked to students' academic achievement. Findings have shown that school location has significant cross-level interaction effects with students' gender in relationship to literacy. That is, for schools located in rural areas, the effect of students' gender on literacy was greater than for schools located in urban areas [34]. School climate also has significant cross-level interaction effects with students' self-concept in relationship to mathematics achievement. For schools with lower school climate, the positive effect of students' self-concept on mathematics achievement was greater than for schools with higher school climate [35]. There was also a significant interaction between school-level SES and the frequency with which each student exhibited problematic behavior in school settings in relationship to reading achievement. School-level SES weakened the negative effect of problematic behavior on reading achievement [36].

## 2.5. Research Questions

The above literature provides a certain reference for us to explore the factors linked to students' academic achievement. However, factors that affect academic achievement vary according to cultural

context. Moreover, the data in educational research is often hierarchical nested data (students nested within schools). For such a data structure, multilevel modeling analysis should be performed to systematically examine the effects of both student- and school-level factors on students' academic achievement. If the hierarchy of data were not taken into account, it is likely that an unreasonable or even wrong explanation of the data would be produced. However, to our knowledge, no multilevel modeling-based research has been conducted to examine factors that affect the academic achievement of ethnic minority students in China. Therefore, this study tried to address the following open questions:

*RQ1*: Is there any significant interschool difference in the academic achievement of ethnic minority students from different schools?
*RQ2*: How are student-level factors linked to academic achievement?
*RQ3*: How are school-level factors linked to academic achievement while factors at a student level are controlled?
*RQ4*: How are the cross-level interactions between student- and school-level factors in relationship to students' academic achievement?

## 3. Materials and Methods

### 3.1. Participants

This research focuses on ethnic minority students from Xinjiang, called Xinjiang Class. Two-stage sampling was used to collect data in this study. The first stage consisted of all 14 provinces and cities that host Xinjiang classes. Then, in the second stage, schools were randomly selected using a ratio of 1:3 for a total of 31 schools. All 12th grade Xinjiang classes' students from the sample schools participated in the questionnaire survey and mathematics test online, with a total of 1873 students. Among them, 664 (35.5%) were boys and 1209 (64.5%) were girls. A total of 972 (51.9%) students were enrolled in the divided class, which means that ethnic minority students studied in the same school as local students, but in separate classrooms, whilst 901 (48.1%) participants were in the mixed class, which means that ethnic minority students studied in the same classroom as local students. In total, 628 (33.5%) students were from urban regions, while 1245 (66.5%) students were from rural regions. In addition, questionnaires were randomly sent to teachers in these 31 sample schools and the number of samples in each school was about 30. A total of 932 effective teacher questionnaires were collected online, including 410 male teachers (44.0%) and 522 female teachers (56.0%). The average age was 40 and the average length of teaching experience was 17 years.

### 3.2. Measures

Because of the importance of mathematics to the career development of minority students and the sustainable development of minority areas in China, the mathematics achievement of ethnic minority students was used as the dependent variable in this study. A mathematics test paper was prepared by the proposition experts in accordance with the "Mathematics Curriculum Standard for Senior High School in China", and in strict accordance with the proposition process. The final test paper consisted of 20 items, including 10 multiple-choice questions, 6 filling-in questions, and 4 problem-solving questions. The full score was 100.

The independent variables at the student level were obtained from students' questionnaires, including gender, class organization, family SES, learning strategy, and learning self-efficacy. Furthermore, the independent variables at the school level were obtained from teachers' questionnaires, including school location, teachers' job satisfaction, and school spirit. Table 1 shows the detailed descriptions.

**Table 1.** Descriptions of the outcome and predicted variables.

| Outcome/Predictors | Descriptions | Mean | SD | Cronbach's $\alpha$ Coefficient |
|---|---|---|---|---|
| | **Academic achievement** | | | |
| Mathematics | — | 69.22 | 18.03 | 0.78, 0.72[1] |
| | **Student-level predictors** | | | |
| Gender | 1 = male (35%), 0 = female (65%) | — | — | — |
| Actual class organization | 1 = divided class (52%), 0 = mixed class (48%) | — | — | — |
| Expected class organization | 1 = divided class (30%), 0 = mixed class (70%) | — | — | — |
| Socioeconomic status (SES) | Based on three indicators: parents' education level, occupation, and family property status [37] | −0.02 | 0.88 | — |
| Learning strategies | Based on six items [38], minimum = 1 and maximum = 5 | 3.69 | 0.59 | 0.80 |
| Learning self-efficacy | Based on five items [39], minimum = 1 and maximum = 5 | 3.75 | 0.58 | 0.78 |
| | **School-level predictors** | | | |
| School location | 1 = urban center (39%), 0 = suburb (61%) | — | — | — |
| Teachers' job satisfaction | Based on eight items [40], minimum = 1 and maximum = 5 | 3.87 | 0.35 | 0.92 |
| School spirit | Based on six items [31], minimum = 1 and maximum = 3 | 2.55 | 0.20 | 0.83 |

Note: 1. The reliability of 10 multiple-choice questions and 6 filling-in questions is 0.78, and 4 problem-solving questions is 0.72.

### 3.3. Data Analysis

The collected data were hierarchical nested data, so two-level modeling was conducted to identify student- and school-level factors linked to mathematics achievement. All variables (except dichotomous variables such as gender, class organization, and school location) were grand-mean centered. The restricted maximum likelihood estimation method was used to estimate the regression coefficient parameters and variances. Multilevel modeling analysis was done in HLM6.0 statistical software. A set-up approach was used to build up the model and the specific steps are as follows [41]:

Step 1: A null model with no predictive variables at both a student and school level was examined to investigate whether there were significant differences in student's mathematics achievement between schools according to the total variance in mathematics achievement accounted for by the student- and school-level predictors;

Step 2: All student-level predictive variables were introduced in the level one equation of the null model, in order to examine the effects of student-level predictors on student's mathematics achievement;

Step 3: All school-level predictive variables were added to the level two equation of the model from Step 2, to determine the effects of school-level predictors on student's mathematics achievement, as well as the cross-level interaction effects between student- and school-level predictors.

## 4. Results

### 4.1. Results for the Null Model

First, in order to test whether the mathematics achievement of ethnic minority students was significantly different between schools, a null model (Model 1) with no predictive variables at both student- and school- levels was investigated. Model 1 is as follows:

Level 1 (student-level) : $y_{ij} = \beta_{0j} + \varepsilon_{ij}, \varepsilon_{ij} \sim N(0, \sigma^2)$,

Level 2 (school-level) : $\beta_{0j} = \gamma_{00} + \mu_{0j}, \mu_{0j} \sim N(0, \tau_{00})$,

where $y_{ij}$ is the mathematics achievement of student $i$ in school $j$; $\beta_{0j}$ is the mean mathematics achievement of school $j$; $\gamma_{00}$ is the grand mean of mathematics achievement across all schools; $\varepsilon_{ij}$ and $\mu_{0j}$ are the random error of the student and school level respectively; and $\sigma^2$ and $\tau_{00}$ are the variations at the two levels, respectively.

Table 2 shows the parameter estimation results of the multilevel modeling analysis. Random variance at the school level was 37.106 ($p < 0.001$), which means that the mathematics achievement of

ethnic minority students was significantly different between schools. The interclass class correlation (ICC) $\rho = \frac{37.106}{(37.106+287.956)} = 11.4\%$, indicating that 11.4% of the total variance in students' mathematics achievement accounted for the difference between schools. According to the judgment standard ($\rho > 5.9\%$) suggested by Cohen [42], this showed that it was not suitable to perform general multiple regression analysis, so multilevel modeling analysis had to be performed to investigate the hierarchical nested data.

**Table 2.** Parameter estimation results obtained by multilevel modeling analysis.

| Parameter | Mode1 | Mode2 | Mode3 |
|:---|:---:|:---:|:---:|
| Fixed effect | | | |
| Student-level predictors | | | |
| $\gamma_{00}$ (intercept) | | 69.02 *** | 68.68 *** |
| $\gamma_{10}$ (gender) | | 2.29 * | 1.33 * |
| $\gamma_{20}$ (SES) | | −0.09 | −0.09 |
| $\gamma_{30}$ (actual class organization) | | 0.01 | 0.60 |
| $\gamma_{40}$ (excepted class organization) | | −2.60 * | −0.49 * |
| $\gamma_{50}$ (learning strategies) | | 1.73 * | 1.68 * |
| $\gamma_{60}$ (learning self-efficacy) | | 6.00 *** | 6.03 *** |
| School-level predictors | | | |
| $\gamma_{01}$ (school spirit) | | | 8.87 |
| $\gamma_{02}$ (job satisfaction) | | | 3.14 * |
| $\gamma_{03}$ (urban center) | | | −0.02 |
| Interaction between student- and school-level predictors [1] | | | |
| $\gamma_{43}$ (urban center and excepted class organization) | | | −4.95 * |
| Random effect variance component | | | |
| $U_0$ | 37.106 *** | 36.42 *** | 29.51*** |
| R | 287.956 | 266.55 | 270.21 |
| Deviance | 15985.42 | 15851.86 | 15803.47 |

Notes: 1. Only variables that have statistically significant interactions are listed. 2. * $p < 0.05$, ** $p < 0.01$, and *** $p < 0.001$.

## 4.2. Results for the Model with Student-Level Predictors Only

All student-level predictors were introduced in the Level 1 equation of Model 1 and were fixed at first. Four predictors—gender, expected class organization, learning strategies, and learning self-efficacy—had significant predictive effects on students' mathematics achievement. Therefore, the predictive effects of these four predictors were considered to be random. Through the chi-square test of the random effect variance components of these variables, only the effects of gender and expected class organization were found to vary across schools, while the error variances of learning strategies and learning self-efficacy were not significant. Therefore, the effects of these two variables (gender and expected class organization) were fixed. The final Model 2 is as follows:

Level 1 (student-level) :

$$Y_{ij} = \beta_{0j} + \beta_{1j}(\text{gender}) + \beta_{2j}(\text{SES}) + \beta_{3j}(\text{actual class organization}) + \beta_{4j}(\text{expected class organization}) + \beta_{5j}(\text{learning strategies}) + \beta_{6j}(\text{learning self-efficacy}) + \varepsilon_{ij}$$

Level 2 (school-level) :

$$\beta_{0j} = \gamma_{00} + \mu_{0j}$$
$$\beta_{1j} = \gamma_{10} + \mu_{1j}$$
$$\beta_{2j} = \gamma_{20}$$
$$\beta_{3j} = \gamma_{30}$$
$$\beta_{4j} = \gamma_{40} + \mu_{4j}$$
$$\beta_{5j} = \gamma_{50}$$
$$\beta_{6j} = \gamma_{60}$$

From the parameter estimation results of the multilevel modeling analysis presented in Table 2, it was found that all student-level indicators had significant predictive effects on students' mathematics achievement, except for family SES. The positive coefficient of gender suggested that boys performed better on mathematics tests than girls. The positive coefficients of learning strategies and learning self-efficacy suggested that students scored higher when they had more self-efficacy in learning mathematics and used more effective mathematics learning strategies. It is also worth noting that learning self-efficacy was the strongest predictor of students' mathematics achievement. With one scale-point increase of learning self-efficacy, students' mathematics achievement average increased by 6 points. Additionally, the negative coefficient of expected class organization suggests that students who expected to be enrolled in a mixed class had a significantly higher mathematics achievement than those who expected to be enrolled in a divided class. Model 2 explained 7.4% of the student-level variance in mathematics achievement.

### 4.3. Results for the Model with Both Student- and School-Level Predictors

All school-level predictors were further added to the final Model 2 to determine the effects of school-level predictors on students' mathematics achievement, after accounting for the student-level predictors. According to the set-up approach [41], these school-level variables were included as predictors for the Level 1 intercept, gender, and expected class organization, which significantly varied across schools in Model 2. The final Model 3 is as follows:

Level 1 (student-level) :

$$Y_{ij} = \beta_{0j} + \beta_{1j}(\text{gender}) + \beta_{2j}(\text{SES}) + \beta_{3j}(\text{actual class organization}) + \beta_{4j}(\text{expected class organization}) + \beta_{5j}(\text{learning strategies}) + \beta_{6j}(\text{learning self-efficacy}) + \varepsilon_{ij}$$

Level 2 (school-level) :

$$\beta_{0j} = \gamma_{00} + \gamma_{01}(\text{school spirit}) + \gamma_{02}(\text{teachers' job satisfaction}) + \gamma_{03}(\text{school location}) + \mu_{0j}$$
$$\beta_{1j} = \gamma_{10} + \gamma_{11}(\text{school spirit}) + \gamma_{12}(\text{teachers' job satisfaction}) + \gamma_{13}(\text{school location})$$
$$\beta_{2j} = \gamma_{20}$$
$$\beta_{3j} = \gamma_{30}$$
$$\beta_{4j} = \gamma_{40} + \gamma_{41}(\text{school spirit}) + \gamma_{42}(\text{teachers' job satisfaction}) + \gamma_{43}(\text{school location})$$
$$\beta_{5j} = \gamma_{50}$$
$$\beta_{6j} = \gamma_{60}$$

From the parameter estimation results presented in Table 2, it was found that only teachers' job satisfaction positively predicted students' mathematics achievement. In schools where teachers had a higher level of job satisfaction, students tended to achieve a higher average score. In addition, although school location had no significant direct effect on students' mathematics achievement, there was a significant interaction between school location and expected class organization. The coefficient of expected class organization was negative in Model 2, and the regression coefficients of urban center and expected class organization were also negative in this model, which implied that school location strengthened the relationship between expected class organization and students' mathematics achievement. For schools located in the urban center, the effect of class organization on students' mathematics achievement tended to be stronger. Model 3 accounted for 6.16% and 20.47% of student- and school-level variance, respectively.

## 5. Discussion and Conclusions

Multilevel modeling analysis was conducted to identify student- and school-related factors linked to the mathematics achievement of minority boarding students in China. The analysis results showed that about 11.4% of the total variability in students' mathematics achievement accounted for the difference between schools. On this basis, multilevel modeling analysis was further

conducted to identify a number of student- and school-level predictors that were significantly linked to mathematics achievement.

### 5.1. Potential Costs and Benefits of Boarding Schools

This special boarding school system has both potential costs and benefits. There are great culture differences between different regions and ethnic groups. From the frontier minority regions to the interior and coastal cities, these young minority students may face various difficulties of sociocultural and psychological adaptation [43]. If these young students cannot adapt to the new cultural environment and as a result have difficulty concentrating on their studies, this could have an adverse impact on academic achievement. In addition to this, other potential costs of boarding schools have been highlighted in the sociological and psychological literature, including a lack of school–family partnership and loss of identity [2].

On the other hand, although such boarding schools adopt closed management, diversified and colorful theme activities for minority students are often carried out, such as visiting national scenic spots and historical museums, which help the ethnic minority students from frontier regions to broaden their horizons [44]. In addition, such boarding schools can potentially provide ethnic minority students with a more complete education, with better educational resources and more professional teachers than those that may be available at their hometown schools. Ethnic minority students could also spend much more time with their local mainstream counterparts, which could intensify "peer effects". In addition to these benefits, there are more potential benefits, such as ensuring students have a healthy lifestyle and spending more time on academic work [2].

Weighing up the pros and cons, we recommend such boarding schools as the preferred senior high education options for remote ethnic minority students. Young minority students in senior high schools already have some independent thinking ability and self-regulation ability, so they can better integrate into mainstream society. Boarding schools at this stage represent the option with the highest cost-benefit ratio.

### 5.2. Student-level Predictors Linked to Students' Mathematics Achievement

The results indicated that gender had a significant predictive effect on students' mathematics achievement. Boys' mathematics achievement was significantly higher than that of girls. This is consistent with the overall situation of senior high school students in China. For example, based on the results of the 2016 Chinese national college entrance examination, studies have found that boys outperformed girls in mathematics [45].

Inconsistent with most previous findings, family SES had no significant effects on students' mathematics achievement. However, this result confirms the findings obtained by Heyneman and Loxley [18]. According to the enrollment policy of interior ethnic boarding schools, 80% of the ethnic minority students should be from remote poor agricultural and pastoral areas [43]. As a result, the family background of these students is mostly similar, and social stratification between groups is not obvious as the vast majority are from poor families, and few are from rich or middle class families, which may weaken the effect of SES on academic achievement.

As for the class organization, the results indicated that expected class organization had a significant predictive effect on students' mathematics achievement. Previous studies have shown that class organization had a great impact on students' learning interest and academic self-concept [46]. Students in mixed classes have always lived and studied with local students, and peer effects and positive student role models would motivate the study of ethnic minority students, especially for the cultivation of good learning habits and learning perseverance. Students who expect to be in mixed classes are usually those with a stronger learning ability and greater learning potential. They are looking forward to accepting more challenges. Students who expect to be enrolled in divided classes are usually less motivated to learn, and their learning self-confidence is easily frustrated, so their academic achievement is relatively lower.

Consistent with previous findings, learning strategies had a significant positive predictive effect on students' mathematics achievement [24,25]. Studies have shown that in basic education teaching in ethnic areas, the professional ability of mathematics teachers is insufficient, and the teaching methods of teachers are mainly based on traditional lecture-style teaching methods. The mathematics class has even become a memory recitation class. As a result, students generally adopt learning strategies of passive listening, mechanical memorization of theorems and formulas, simple imitation and exercises, and a lack of interaction and communication [17]. However, in the stage of senior high school, with the increase of mathematics learning content, the difficulty of learning also increases. In this case, how to study mathematics efficiently becomes very important to mathematics achievement. If students continue to use previous learning strategies, it will be difficult to achieve good learning results.

Consistent with previous findings, learning self-efficacy had significant positive predictive effects on students' mathematics achievement [26,27]. It is worth noting that learning self-efficacy was the strongest predictor of students' mathematics achievement. Studies have shown that there is a big-fish–little-pond effect in learning self-efficacy [28,29]. Such boarding schools are all set in the provincial model high schools with the best educational resources, most professional teachers, and excellent local students. Although these minority students are excellent junior high school graduates selected in minority areas through a unified admission examination, there is still a big academic gap between them and local students. The fierce competition in learning and the psychological gap will affect the self-confidence and learning self-efficacy of minority students. If they cannot actively adjust their mentality in time, it will further affect their academic achievement.

### 5.3. School-Level Predictors Linked to Students' Mathematics Achievement

Inconsistent with previous findings, school ethos had no significant effect on students' mathematics achievement [31]. This is mainly because such boarding schools are set in provincial demonstration high schools with high-quality education in developed cities. Teacher's behavioral norms, professional ethics, students' learning discipline, as well as the overall school spirit and style of study, are all good.

Consistent with previous findings, teachers' job satisfaction had a significant predictive effect on students' mathematics achievement [32,33]. Compared with teachers in ordinary schools, teachers in such boarding schools face more challenges in teaching and management. For example, these boarding ethnic minority students only go home once a year during the summer vacation, which makes the work content of teachers more complicated and cumbersome, and the work intensity and workload are very heavy. Overtime work becomes the norm. In addition, the absence of parents' support also brings difficulties to teaching and management. Moreover, the learning foundation and ways of thinking of these students are quite different from their mainstream counterparts in developed cities, and these factors also increase the difficulty of teachers' teaching. Additionally, the lack of social understanding of such boarding schools leads to little social support and low recognition for teachers. Teachers' job satisfaction is easily affected by these factors. As one of the key factors in education and teaching, teachers' job satisfaction plays an important role in the improvement of students' academic achievement.

School location has significant cross-level interaction effects with expected class organization in relationship to mathematics achievement. Regression results revealed that for schools located in the urban center, the effect of students' expected class organization on mathematics achievement was greater than schools located in the suburbs. There are certain differences in the management of boarding schools due to different school locations [18]. The surrounding environment of schools in the urban core area is more complicated, so schools basically implement fully enclosed strict management. Students rarely have the opportunity to communicate with outside schools. In the suburbs, the surrounding environment is relatively simple, so the management of schools is more open and relaxed, and students are freer to go out. This also relieves students' learning tension to a certain extent and help students better integrate into the inland environment and mixed-class teaching.

*5.4. Recommendations for Education Policy Making*

Based on the above findings from multilevel modeling analysis, the researchers put forward the following discussion and suggestions for the sustainable development of minority education in China.

(1) Mixed-class teaching should be further promoted.

According to the students' questionnaire data, the proportion of minority students who were actually in a mixed class was 48%, but the proportion of students who expected to be in a mixed class was up to 70%, which indicates that mixed-class organization represents the aspiration of most students. Moreover, the results of multilevel modeling analysis showed that students who expected to be enrolled in a mixed class had significantly higher mathematics scores than those who expected to be enrolled in a divided class. Ethnic minority students in a mixed class have lived and studied with local students, which can help these boarding students to integrate into the local study and life as quickly as possible. In this way, mixed-class teaching will not only contribute to the improvement of students' academic achievement, but also contribute to the interethnic communication among students of different nationalities.

(2) The interior ethnic boarding schools should be set in the suburbs.

The results of multilevel modeling analysis highlighted that school location positively predicted the effect of expected class organization on students' mathematics achievement. For schools located in the urban center, the effect of class organization on students' mathematics achievement was greater than schools located in the suburbs; that is, for schools which are located in the urban center, students who expect to be in a divided class tend to achieve a lower average score. Therefore, we suggest that interior ethnic boarding schools should be set in suburb schools that have a good school spirit, simple surrounding environment, and medium academic achievement. In these school environments, ethnic minority students display a small difference in academic achievement from local students, so they are more likely to build up confidence and have a stronger sense of self-efficacy in learning. It would therefore be helpful to promote mixed-class teaching in such an environment. The surrounding environment of suburb schools is also relatively simple, and schools' management is more open and loose, which also alleviates the learning tension of ethnic minority students to some extent and helps them better integrate into the new environment, thus contributing to the improvement of their academic achievement.

(3) Teachers' job satisfaction should be improved through multiple measures.

The results of multilevel modeling analysis also showed that teachers' job satisfaction was significantly linked to students' academic achievement. Compared with teachers in ordinary schools, teachers in interior ethnic boarding schools face more challenges in teaching and management, so more attention should be paid to such teachers' job satisfaction. Therefore, we suggest (a) improving the salary of teachers and staff in interior ethnic boarding schools and thus reducing troubles caused by economic problems, in an effort to make them feel at ease teaching in such working environments; (b) that education departments, especially ethnic minority education departments, should allocate special funds or set up special awards to encourage teachers in interior ethnic boarding schools to actively carry out teaching and research activities from the perspective of culture and psychology; and (c) that government departments should increase the publicity of interior ethnic boarding schools, so that teachers can get effective social support and social trust and improve their professional identity and job satisfaction.

## 6. Limitations

By performing two-level modeling analysis, student- and school-related factors linked to students' mathematics achievement were identified. However, the mathematics scores used here are based on the results of only once test: a summative assessment. Information obtained by this single evaluation method may not be accurate. For a more scientific and accurate evaluation of the factors influencing students' academic achievement, value-added assessment should be considered in further research [47,48]. Value-added assessment refers to collecting standardized test scores of students at

different time points within a specified period of time, through tracking research design. Based on a longitudinal comparison of students' own test scores, multilevel modeling could be used for statistical analysis of the data to track students' academic changes over a period of time, and to examine the net effects of school- or student-level predictors on students' academic achievement, so as to achieve scientific and objective assessment results.

**Author Contributions:** A.S. proposed the present idea of this study, designed the research program, collected the raw data, processed the data using software, and wrote the original draft; W.H. collected the relevant policy documents, and reviewed and edited the manuscript. All authors have read and agreed to the published version of the manuscript.

**Funding:** This research was funded by the National Social Science Fund of China, grant number CMA150130.

**Conflicts of Interest:** The authors declare no conflicts of interest.

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
