# Peer review of "Exploring Factors Linked to the Mathematics Achievement of Ethnic Minority Students in China for Sustainable Development: A Multilevel Modeling Analysis"

_sustainability, doi:10.3390/su12072755_

Round 1

Reviewer 1 Report

Thank you for making substantial editing.

It is workable to reclassify the school location. Only one thing, please keep in mind to set the school location as a categorical in your analytical software. If you already did, it is fine; if not, you may want to re-run it see whether the estimate might change, if you set it as a categorical variable.

Also for reporting the estimates of school location and its interaction, it is better to rename this variable as " urban center" (I figure out you might use "1" for urban center from table 1, and for the rest of the analyses), then the suburban is your reference group, since if you only list school location, readers can not figure out the estimate is related to which school location. 

Author Response

Point 1: It is workable to reclassify the school location. Only one thing, please keep in mind to set the school location as a categorical in your analytical software. If you already did, it is fine; if not, you may want to re-run it see whether the estimate might change, if you set it as a categorical variable.

Response 1: Thank you very much for your kindly suggestion. We have set the school location as a categorical in the analytical software. The parameter estimation results in table 2 are already the results after rerunning the data.

Point 2: Also for reporting the estimates of school location and its interaction, it is better to rename this variable as " urban center" (I figure out you might use "1" for urban center from table 1, and for the rest of the analyses), then the suburban is your reference group, since if you only list school location, readers can not figure out the estimate is related to which school location. 

Response 2: Thank you very much for your kindly suggestion. As gender, class organization and school location are all the dichotomous variables, we added note 2 below table 2 in the article. The specific contents are as follows.

Note 2. The regression coefficients of dichotomous variables (X=1 or X=0) represent the average change in student achievement caused by X=1 (compared to X=0).

Reviewer 2 Report

In my first report (minor revisions) I had suggested that more emphasis should be given to the description of the differences of the learning skills in mathematics among the local and the foreign students in China. I think that the authors have responded satisfactorily to my suggestion in the revised manuscript.

Author Response

We are truly grateful to your critical comments and thoughtful suggestions.